# Is a Purloined Letter Just Writing? Burrowing in the Lacan-Derrida Archive

**Jean-Michel Rabaté**

Department of English, University of Pennsylvania, Philadelphia, PA 19104, USA; jmrabate@english.upenn.edu

**Abstract:** Starting from a recent book on Derrida and psychoanalysis, I return to the controversy between Lacan and Derrida in the 1970s. Its focus was the letter as interpreted by Lacan in a commentary of Poe's "Purloined Letter". While agreeing with some of Derrida's objections, I conclude that Lacan makes stronger points about the destination of the letter. I give my own example, Kafka's "Letter to the Father" in order to argue that one can state that "a letter always reaches its destination" even if it has not been delivered.

**Keywords:** deconstruction; signifier; letter; writing; destination; psychoanalysis; philosophy





The context of my investigation has been shaped by pedagogical experience. Teaching texts by Jacques Lacan, I often hear bright students ask: "Isn't this like deconstruction?" When I express my surprise, each time I hear something like: "Well, you know, the idea that language shapes reality, that you can use it to criticize everything". Half a century after the notorious and acrimonious debates between Lacan and Jacques Derrida, one can expect a degree of blurring and overlapping. Both Derrida and Lacan highlighted the role of language in its ability to shape the world. Truly, Derrida's early motto that there would be no "outside the text" can be compared with Lacan's insistence that the unconscious is structured like a language.

However, given the fact that both Lacanian psychoanalysis and Derrida's analyses recently regained favor among advanced students, to the point that one can talk of a "come back" of Freud in American culture[1] and of a resurgence of deconstruction often presented as the hidden source of the recent spate of "woke" critiques, it is important to avoid damaging confusion. If Derrida can be said to have introduced a certain version of psychoanalysis into philosophy, he never spared Lacan, for once, in his wholesale demolition of certain Freudian myths and pieties. It is, therefore, crucial to historicize the conversation not simply with the idea of deciding who was right or who won, but in order to assess where we stand today when we teach these classical texts. If Derrida and Lacan share a playful and productive usage of the signifier as the material side of language, can one go on to assert that the Lacanian concept of the "letter" is, if not identical, at least compatible with Derrida's deployment of the "text"?

Paul Earlie's wonderful recent book, *Derrida and the Legacy of Psychoanalysis* (Earlie 2021) brought me back to the debate between Lacan and Derrida, a controversy in which my generation found itself caught. In the late sixties, it looked as if one could be both a Derridian and a Lacanian. The late Philippe Sollers[2], Julia Kristeva, and Jacques Alain Miller were on both sides until a split appeared in the seventies. Then, one had to choose one's camp. For a long time, I managed to sit on the fence by using a simple device: I would be Derridean with the Lacanians and Lacanian with the Derrideans, which entailed minimal intellectual contortions. However, at times, one has to prove one's allegiance, but in fact, to my relief, I found that the requirement to toe party lines was limited to France, and I had moved to the US, where boundaries were murkier. When I re-read today the main texts of the controversy that somehow allegorize a clash between philosophy and psychoanalysis, I notice that a crucial concept in that discussion had been the term "letter",

a term that can be taken literally and points to writing, or as referring to messages sent out and in circulation. This double meaning comes to the fore in Lacan's reading of the Purloined Letter, the introductory essay that opens *Ecrits*.

In his celebrated analysis "Seminar on The Purloined Letter", Lacan imposes on Poe's story a structuralist grid that maps out a pattern of displacements: by a series of permutations, each actor in the story stands for a distinct subject position. The subjects are subjects of unconscious desire determined by the Unconscious or discourse of the Other. They all follow the same sequence of actions. The structure achieves its effects by pushing along pure signifiers, in this case, a letter that has been opened and read, then hidden, but whose contents are not disclosed; the letter allegorizes the itinerary of a signifier whose signifieds remain inaccessible and, to a certain extent, irrelevant. What counted most for Lacan were the places characters occupied and the way they were caught up in a repetition automatism; the consequence was that the letter always reached its destination in the end. In his violent polemical readings of the seventies, Derrida rejected such an economy: Lacan would find at the end what he had hidden in plain sight, the letter as the phallus, which would not be so far from a previous reading of Poe's story already presented, albeit in a less sophisticated manner, by Marie Bonaparte in her book on Poe. The proximity Derrida discovered is all the more ironic given the scorn Lacan pours on Bonaparte's heavy-handed psycho-biography. "Le facteur de la vérité", an essay published first in *Poétique*, then reprinted in *The Post Card*,[3] develops a critique of psychoanalysis—or more to the point, an attack on Lacanian theory—Derrida had begun to sketch as early as 1972 in *Positions*. In that book, Lacan was taken to task for his glib use of Hegelian categories and for an undeclared idealism at the same time as he was flaunting the "materiality of the signifier".

In 1975, Derrida added new reproaches. Lacan was accused of simplifying Poe's text and missing literary nuances, glossing over the intertextual plays elaborated in the Dupin stories of detection. The main objection was that Lacan translated the absent content of the letter into a Freudian "truth". This truth, according to Lacan, would be identical to the truth of psychoanalysis; it would reveal the phallus that had been hidden in the empty signifier of the letter. The letter then ultimately is a sign of castration.

Derrida also categorically refused Lacan's assertion that "a letter always reaches its destination". As he noted, letters can always get lost, stolen, or burned and destroyed. Derrida rejected an all-too-ideal predestination. He then played with it in *The Post Card*, a tricky text in which he multiplied performatives, or rather "pervert-formatives", (Cotton 2019) while using the form of the love letter. "Envois" is thus made up of fragments remaining from an amorous correspondence that he had decided to destroy.

Indeed, Lacan's argument cannot be dissociated from his analysis of the nature of letters. For Lacan, a letter remains a single entity, no matter what happens to it. A letter is said to be "uncuttable". In French, if one can speak of "letters" in the plural, one cannot say: "there is letter", or "we have here some letter", as one might say "there is time" or "there is butter here". In the sense of missive, a letter, be it singular or plural, cannot be counted or divided. Even if it was cut materially, the fragments would still make up one letter. This is why, in Poe's plot, the police are mistaken in their assumption that the Minister's room must be divided into smaller units, objects, books, frames, table legs, etc. They repeat Zeno's paradox: if a line is endlessly divisible, then movement is impossible, which means that intelligence dies (Zizek 1991 for a Lacanian discussion of Zeno).

Similarly, a "letter" in French and English is a term predicated on homophony that does not work in German, where we have to distinguish between briefs and "letters" as written signs (Buchstaben) that may be used to write a brief. Even if a letter has been lost or destroyed, it remains present in its absence as a letter. Lacan refuses to distinguish these two meanings. The action of the police proves a contrary that the indivisibility of the letter creates its invisibility. Since their categories cannot assimilate the idea of a reversed and re-signed paper, for them the letter manque à sa place [is missing from its designated location], the expression by which librarians signal that a book has been lost as soon as it has been misplaced. Seeing only what they can divide into smaller units, they miss the

letter spread out in front of their eyes. I would argue that a similar process of division is at work in Derrida's reading, and it can lead him to similar moments of blindness.

Derrida goes further in his critique of Lacan's gesture: "It is against this possible loss that the statement of the 'materiality of the signifier', that is, about the signifier's indivisible singularity, is constructed. This 'materiality', deduced from an indivisibility found nowhere, in fact, corresponds to an idealization. Only the ideality of a letter resists destructive division". ([Muller and Richardson 1988](#), PP, p. 194) The argument that the alleged "indivisibility" is "found nowhere" repeats the gesture of those who blindly look for something and do not find it. Placing himself strategically on the terrain of empirical evidence, arguing that one can always tear up a letter into small pieces, Derrida may be blind to the common enough paradox that the materiality of a single letter does not entail that it is being present. Here, we encounter the vexing issue of the inevitable process of "idealization" performed by whoever goes beyond the pure materiality of the signifier in order to produce a concept.

Poe had anticipated this move when adding an epigraph cited in both German and an imperfect English translation to the second Dupin tale, "The Mystery of Marie Rogêt". The latter is given as: "There are ideal series of events which run parallel with the real ones. They rarely coincide. Men and circumstances generally modify the ideal train of events, so that it seems imperfect, and its consequences are equally imperfect. Thus with the Reformation; instead of Protestantism came Lutheranism".[4] The concept at hand is not the translation of the letter into its meaning (as the phallus or as castration), but it is simply the concept of the Letter—the allegory of the Letter if you want.

Derrida's objections to the letter's economy as a circuit determined by an ideal teleology are more damaging. Indeed, why should a letter "always" return to its original place? Is Lacan implying that all mislaid letters end up in their rightful owners' hands? An answer to this query is found as soon as I relate to the letter in some way: for then, I am the answer, which boils down to stating that any readers reading the "letters" of literature will be their addressees. Lacan's position makes better sense if we add to all his pronouncements the qualification of "in psychoanalysis", or, in this case, "in literature". Slavoj Zizek has made that type of argument in *Enjoy Your Symptom!* ([Zizek 1992](#), Hereafter EYS and page number) Against Derrida's accusations of idealism and teleology, Zizek argues that Lacan's formula functioned at different levels defined by the registers of the Real, the Imaginary, and the Symbolic. At an imaginary level, the tag according to which a letter "always reaches its destination" can be glossed as "its destination is wherever it arrives" (EYS, p. 10). For Lacan, the letter has already arrived in so far as it is understood to be a letter. In order for someone to talk about the letter, the letter must have had at least one recipient; it does not matter if this recipient was not the original addressee. At a symbolic level, the circuit of the letter is an economy, but not a meta-language like a theory of fate. Fate is entirely contained within the letter itself, whether one sees its possession as tragic, gendered, or a blessing. At a Real level, the letter has death as its hidden message. This lies in its very materiality; what could be more "dead" than these inert little signs of a piece of paper, as Derrida himself has often stressed? The letter becomes in Zizek's reading a "dangerous supplement", and it has to be put to rest. It points to the excess of enjoyment, a jouissance that has no name and no function (EYS, p. 22).

A more recent analysis has been proposed by Shirley Sharon-Zisser in a 2020 essay on Derrida's and Lacan's parallel readings of Shakespeare: "The question of the letter and its (non-)arrival is a well-known point of difference between the teachings of Derrida and of Lacan". Lacan argues that a "letter always reaches its destination" ([Lacan 2006](#), p. 30). "Letter" in this case means a series of unconscious signifiers that together generate a grid, a knot, defining something like a line of fate. For any subject, these signifiers remain unknown, but one can show retroactively that they limned the contours of a specific destiny. In *The Post Card*, Derrida responds to Lacan's statement by affirming that "a letter does not always arrive at its destination, and from the moment that this possibility belongs to its structure one can say that it never truly arrives, that when it does arrive

its capacity not to arrive torments it with an eternal drifting" (1987c, 489). While it is of course not possible to disagree with Derrida regarding the letter as a human phenomenon, metapsychologically speaking, it is not tenable to assume that a letter, as Lacan speaks of it, a letter, as an unconscious signifier, would not in some way affect the subject's life (arrive at its destination in Lacan's terms). The possibility of not arriving that, as Derrida says, is structurally endemic to the letter was used in psychoanalytic thinking at its inception—namely, in Freud's early essays on "The Neuro-Psychoses of Defense", where the different neurotic modes of treating an unpleasurable representation as unknown even when it is registered as unconscious by "robbing it of its affect" is referred to by Freud as its being "non-arrivée":

> "As regards the path which leads from the patient's effort of will to the onset of the neurotic symptom, I have formed an opinion which may be expressed, in current psychological abstractions, somewhat as follows: The task which the ego, in its defensive attitude, sets itself of treating the incompatible idea as 'non arrivée' simply cannot be fulfilled by it. Both the memory-trace and the affect which is attached to the idea are there once and for all and cannot be eradicated". (Freud 1962)

This is the basis for Shirley Sharon-Zisser's work on what she calls "Unconscious Grammatology" (Sharon-Zisser 2020, 2022; Freud 1962). If the trajectory defining such a "fate" works at several levels, the gist of the tale is exemplified in Lacan's deliberate transformation of the lines from Crébillon that Dupin copies in his substitute letter. At the end of his essay, Lacan deliberately replaces the word dessin by destin: "... Un destin si funeste,//S'il n'est digne d'Atrée, est digne de Thyeste " (E, p. 40, and PP, p. 52).

A change of one letter transforms the "design" (dessin) of teleology into a "fate" (destin) determined by repetition. This is the repetition that the new letter written by Dupin and now in possession of the minister, unaware of the substitution, holds in store. As Barbara Johnson points out, Crébillon's play revolves around a letter informing King Atreus of his betrayal by his brother Thyestes. This letter, disclosing that his son is the son of Thyestes, triggers the usual roll call of incest, parricide, and even cannibalism (PP, pp. 235–36). Johnson has shown that Derrida tried to exceed the frame provided by Lacan but failed to address other parts of the Seminar on Poe, like the mathematical discussion at the end. Poe had reproduced the mere structure of historical novels whose grid he laid bare by sketching a textual logic of exchanges; the fact that we do not know anything about any content generates all sorts of fantasies about them. In a parallel fashion, Lacan had followed the trajectory of a letter that cannot be sent back to an empirical sender but always reaches its destination because the destination is the locus of the Other, the birthplace of all desires and fantasies. Thus, the unconscious "work" of a letter that moves around in Poe's tale presupposes the ideality of a certain logic, as Derrida contends; however, this circuit is determined by a constant displacement of identities given the slippery nature of all signifiers.

If the Other is more than a word hoard containing all the signifiers and can be grasped as a site reached by letters, we are back to Freud's notion of an Unconscious made up of letters. As Earlie has shown, Derrida and Lacan meet upon the idea that Freud describes the Unconscious as an "arche-writing". It is on this basis that the bifurcation between oral speech and writing will be established. For Freud, the Unconscious is fundamentally a type of writing that can be illustrated by letters, hence, the Greek letters he chose to distinguish between the Phi and the Psi neurons. This type of unconscious writing is also allegorized by literature; it was Poe's genius to have sensed it.

To assess the validity of the trope of the "letter that always reaches its destination", I will take one well-known example, the famous letter that was never really delivered to its addressee, Kafka's "Letter to His Father". Derrida's analysis has probed the perverse strategy of Kafka who gave the letter to his mother, asking her to pass it on to his father. After she had read it, she did not have the courage to give it to her husband. Finally, it was Ottla, Kafka's favorite sister, who ended up giving it back to Franz. The letter,

which was found in the archive, has become a staple of biographical criticism. It has fed countless psychoanalytical readings that all rehearse how Kafka felt awed by his father's strength and domineering manners, chose writing against the father's law and urge to be a businessman, marry, and so on. However, when Derrida looked at the text more closely, he saw a more complex scene and highlighted one of the surprising twists of this text, the moment when, close to the end, Kafka has his father intervene so as to justify himself. Indeed, Kafka's father is made to reply in advance to the ferocious attacks contained in the previous pages. The father is granted two pages that are inserted between quotation marks. Here is Derrida's astute assessment of this vertiginous multiplication of perspectives:

> Extraordinary speculation. Bottomless specularity. The son speaks to himself. He speaks to himself in the name of the father. He causes his father to speak, taking his place and his voice, at the same time lending him and giving him speech: you take me for the aggressor but I am innocent, you attribute sovereignty to yourself in forgiving me, therefore in asking you forgiveness in my place, then in according to me forgiveness and, doing this, you achieve the double blow, the triple blow, of accusing me, of forgiving me, and of acquitting me, in order to finish by thinking me innocent there where you have done everything to accuse me, insisting besides on *my* innocence, and therefore yours since you identify yourself with me. But here is what the father recalls to mind, in truth the law of the father speaking through the mouth of the son speaking through the mouth of the father: if one cannot forgive without identification with the guilty party, one can no longer forgive and excuse *[innocenter]* at the same time (Derrida 2005, (hereafter, LS)).

We can see how Kafka's father incriminates the son's lack of responsibility and sincerity, and he opposes the virile man-to-man fight he is used to with the cunning and devious strategies used by his son Franz. The main site of cunning, perversion, and lack of sincerity has to do with literature, which becomes very clear when we find Kafka evoking the famous beginning of "The Metamorphosis" in which the hero, Gregor, has been changed into a monstrous vermin (Ungeziefer). In the "Letter", the father tells the son:

> And there is the combat of vermin (des Ungeziefers), which not only sting but, on top of it, suck your blood in order to sustain their own life. That's what the real professional soldier is, and that's what you are. You are unfit for life; to make life comfortable for yourself, without worries and without self-reproaches, you prove that I have taken your fitness for life away from you and put it in my own pocket. Why should it bother you that you are unfit for life, since I have the responsibility for it... (Kafka 1989)

Readers of Kafka will have recognized not only the key term of "The Metamorphosis" ("vermin") but also the plot of "The Judgment", and, what is more, the narrative structure of "Eleven Sons", a story in which Kafka takes the stern attitude of his father to describe, one after the other, eleven of his own sons. The trick is that each son corresponds to one of his short stories. Indeed, Kafka saw his texts as his "sons", but in that particular story, each time the father pretends to praise a son it is in order to find a damaging defect in the qualities he lists. When Derrida pursues his analysis, he quite rightly identifies literature with a form of filial parasitism:

> We will not comment on the end of this letter to the son, a fictive moment of the also entirely fictive *Letter to his Father*. But at the bottom, it carries in itself, perhaps, what is essential in this secret passage from the secret to literature as an aporia of forgiveness. The accusation that the fictive father will never withdraw, the grievance that he will never symmetrize or specularize (by the fictive voice of the son, according to that legal fictio that, like paternity according to Joyce, defines literature), is the accusation of parasitism. It runs through the whole length of the letter, of the fiction, and of the fiction in the fiction. It is, finally, literary writing itself that the father accuses of parasitism (LS).

Kafka's letter has thus at least two addressees, his father and himself. Derrida formulates this doubleness clearly:

> What, in its destinal trajectory, was the letter of the father inscribed in the letter to the father, of Kafka? In the letter of the father of Kafka to the son and signatory of the letter to the father of Kafka, across all the genitives and all the signatures of this forgiving genealogy? Irrecusably, this letter of the father to the son was also a letter of the son to the father and of the son to the son, a letter to himself of which the stakes remained that of forgiveness of the other that was forgiveness of the self. Fictive, literary, secret but not necessarily private, it remained, without remaining, between the son and himself. But sealed in his inmost heart, in secret, in the writing desk (secretaire) in any case, of a son who writes to himself in order to exchange without exchanging this abyssal forgiveness with the one who *is* his father (who becomes in truth his father and bears that name after this incredible scene of forgiveness), this secret letter becomes literature, in the literality of its letter, only at the moment when it is put on display to become a public and publishable thing, an archive to inherit, still a phenomenon of inheritance—or a last will and testament that Kafka will not have destroyed (LS).

Assuredly, the mistake of most psychoanalytic readings was to assume that the true addressee of the letter was Kafka's mother. She was indeed the first one to read it, but the danger is to reduce the spiral of such endless mirror identifications to an Oedipal pattern that would conclude that this letter aimed at killing Kafka's father symbolically and reclaiming his mother. In fact, the letter does not put the mother in a different category: she is not excluded from the reproaches voiced, because she appeared also tainted by her complicity with the father's ideology. It remains true that the would-be bride, the young woman whose courtship was the main occasion for the letter, Julie Wohryzek, shared a first name with Kafka's mother, Julie Kafka.

The issue underpinning the letter is Kafka's inability to marry, first because his father objected to his marriage with Julie, and then because the very idea of marriage would prevent him from writing. Kafka felt that he had been forever married to literature. The task of penning the 66 manuscript pages, taking a break just for the writing task, had a successful outcome. Kafka would explain his sudden decision not to marry the family of Julie and overcome a writer's block that had stymied him for one year. From that moment until his death, his productivity was frantic.

Hence, to read these texts adequately, we do need categories like the Other (Kafka's letter was addressed to his symbolic father, not to his real father) along with a dialectic of idealities underpinning the curious trajectory of that letter. Having written the undelivered letter to his Father, Kafka managed to deliver a letter—that actually was partly lost for a while—to Julie Wohryzek's sister. He explained in detail why he could not marry Julie and took his father's opposition to the marriage as proof that he was right to consider it! This zigzag allowed him to analyze the decision to marry from the point of view of the Other. Only then could he sort out priorities, and of course the priority was neither sexual happiness nor the possibility of marrying and having children but literature. If this seems to confirm Lacan's thesis, Derrida has a point: we have lost the correspondence between Kafka and Julie, but precisely because these letters have disappeared, we cannot talk about their destination. They remain as a gap in the convoluted arabesques of Kafka's desire. Derrida was reading the letter to the father as haunted by the story of Abraham and Isaac. We know that Kafka was an avid reader of Kierkegaard and that he tried to reply to Kierkegaard's fascinating analysis of the theme of sacrifice, Abraham's silence, and Isaac's terror. Thus, Derrida writes:

> For, as in the sacrifice of Isaac which was without witness or had no surviving witness except the son, that is, a chosen heir who will have seen the tense face of his father at the moment when he lifted the knife above him, all of that happens to us only in the trace left by the inheritance, a trace left legible as much as illegible. This trace left behind, this legacy, was also, by calculation or by unconscious

imprudence, the hazard or the risk of becoming a testamentary word in a literary corpus, becoming literary by this very abandon. This abandon is itself abandoned to its drift by undecidability, and thus by the secret, by the destinerrance of the origin and the end, of the destination and the addressee *[destinataire]*, of the meaning and of the referent of the reference that remains reference in its very suspense. All this belongs to a literary corpus as undecidable as the signature of the son and/or of the father, as undecidable as the voices and the acts that are there exchanged without exchanging anything (the "true" father of Kafka, no more than Abraham, has perhaps understood nothing and received nothing and heard nothing from the son... (LS).

Derrida is right to highlight the undecidable, which leaves us with a final hesitation, whose constant reversals become more poignant when we discover the ironic treatment of the Abraham/Isaac paradigm in Kafka's later letters to his friend Robert Klopstock. Kafka seems to mock both Kierkegaard and the Biblical tradition when he states that he could imagine "another Abraham" who would not have answered the call because he would always have something do to, and then again "another Abraham" who could not accept the divine order because he thinks he has been called by mistake, like a bad student who thinks he has heard his name called for a final distinction, but of course he is wrong (Kafka 1977).

The second point brought home to me by *Derrida and the Legacy of Psychoanalysis* is an insight into the constancy of Derrida's program. Derrida's main insights arrived early, and he never deviated from them. Derrida's reading of Freud in "Freud and the Scene of Writing" harked back to a student paper written for Louis Althusser, "The Unconscious", from 1954–55. This was followed and expanded by two 1970 seminars on psychoanalysis that are often quoted by Earlie. Derrida's program was elaborated early on, and he remained faithful to it, which includes *Archive Fever* and the conversations about psychoanalysis with Elizabeth Roudinesco. In 2001, Derrida still wondered whether Freudian concepts like the ego, the super-ego, primary processes, repression, etc., had not become totally obsolete. They had been "provisional weapons" built slap-dash by Freud in order to attack philosophies of consciousness, and they have no future (Derrida and Roudinesco 2003).

In the first approaches, Derrida seemed more optimistic, for he saw in Freud's writing machines the chance of eschewing "metaphysical thinking" thanks to their mixing up life and death in the psyche of individuals. Earlie even stresses a certain anxiety of influence in Derrida when he refused the idea that he would have applied Freud's concept of the Unconscious to philosophy, a rejection he voiced more than once. In *Writing and Difference*, one can nevertheless admit that there is a Freudian concept of the trace that corresponds to Derrida's writing. Once more, it is the way it gives a place to death that prevents Freud from falling into a metaphysical notion of a biological "life".

Freud's decisive confrontation with death took place during WWI. In 1920, he wrote *Beyond the Pleasure Principle*, a book in which he explores notions of repetition, pleasure, and reality taken as three fundamental principles. In chapter five, we see Freud generalizing and moving from the compulsion to repeat to the "natural" tendency he finds everywhere, a tendency to return to a previous state of things, in other words, to a principle of entropy. Life would just be a detour, a "circuitous path" along a return to inorganic matter from which it has come. This could then be rephrased via the old motto: "The aim of all life is death" (Freud 1989, Hereafter, BPP and page number). A whole page develops this idea, with the striking image of the "guardians of life" who helped the organism strive for survival and then transformed into the "myrmidons of death" (Freud 1982, and BPP, p. 47).

When Derrida comments on Freud's text in the *Post Card*, he mentions Heidegger and suggests that *Beyond the Pleasure Principle* and *Being and Time* have similar programs:

When Freud speaks of *Todestrieb, Todesziel, Umwege zum Tode*, and even of an "*eigenen Todesweg des Organismus*", he is indeed pronouncing the law of life-death as the law of the proper. Life and death are opposed only in order to serve it. Beyond all oppositions, without any possible identification or synthesis, it is indeed a question of an economy of death, of a law of the proper (oikos, oikonomia), which governs the detour and indefatigably seeks

the proper event, its own, proper propriation (Ereignis) rather than life and death (Derrida 1987b, Hereafter, P and page number).

Derrida perceives a crucial knot, although the conflation of Heidegger and Freud is misleading, and makes him miss the progression of Freud's text. This is surprising given the slowness of his previous commentaries of *Beyond the Pleasure Principle*. Derrida stops reading at some point and does not see that the thesis of "all life is bound to end in death" is later taken as a philosophical cliché that Freud rejects. Freud turns around and exclaims: "It cannot be so" (BPP, p. 47). One would have to follow in detail the complex progression from the end of chapter V to chapter VI to see Freud multiplying aporias and counter-examples.

This is a key passage because of how the death drive is introduced. The phrase appears in parentheses in the original text: "The opposition between the ego or death instincts and the sexual or life instincts would then cease to hold..." (BPP, p. 53, "*Der Gegensatz von Ich(Todes-)trieben und Sexual(Lebens)trieben würde dann entfall*en..." (*Jenseits*, p. 253)). We read this surprising assertion: "Let us turn back, then, to one of the assumptions that we have already made, with the exception that we shall be able to give it a categorical denial" (BPP, p. 53, *Jenseits*, p. 253). Freud then debunks his previous assertion that all life and nature are moving inexorably to death. The idea is rejected because Freud thinks that it is too comforting: "Perhaps we have adopted the belief because there is some comfort in it. If we are to die ourselves, and first to lose in death those who are dearest to us, it is easier to submit to a remorseless law of nature...". (Freud 1989, p. 53). Freud is telling us that if entropy can be construed as a "law of Nature", we have not voted this law, to paraphrase Joyce's quip in *Exiles*.[5]

Freud reopens the biological debate and contrasts the dying cells of any organism with an undying germplasm. Death becomes less "natural" if it appears as a later acquisition of organisms. Woodruff had shown that infusorians can, if placed in a refreshed environment that nourishes them, reproduce themselves by fission for more than 30,000 generations (BPP, p. 57). In cases when the solution has not been renewed and degeneration is observed, this process can be reversed if two animalcules blend together; they achieve regeneration and avoid the degeneration that leads to death. It is in the context of these speculations that Freud asserts that he believes in a dualism of the drives, a principle that is constructive (aufbauend) and a principle that is "de-structive" or "deconstructive" (abbauend) (BPP, p. 59, *Jenseits*, p. 258).

Derrida had objected to such a translation earlier in his chapter and criticized readers who would retroactively import "deconstructive" themes in translations of Marx and Nietzsche. Finally, he grudgingly accepts that one might engage in such a project although he betrays some ambivalence: "If one were to translate abbauen as "to deconstruct" in *Beyond*... perhaps one would get a glimpse of a necessary place of articulation between what is involved in the form of an athetic writing and what has interested me up to now under the heading of deconstruction" (P, p. 268). Why this resistance? Is it that he is afraid that Freud had anticipated the issue of "un-building" or done this better than Heidegger?

In these introductory pages to a long and detailed reading of Freud's *Beyond the Pleasure Principle* (Freud's text is less than 80 pages whereas Derrida's commentary is more than 150 pages long), Derrida discusses Freud's debts to philosophy, insisting rather on his denial of any debt. Derrida lists Schopenhauer and Nietzsche above all, noting indeed the curious fact that as soon as Freud seems to agree with Schopenhauer, he takes his "bold step forward". Derrida even quotes the text in German (P, p. 268) before describing Freud's strategy in startling terms: it does not come back to itself in a Hegelian manner, it does not follow a hermeneutic circle but it progresses according to a series of detours: "It constructs–deconstructs itself according to an interminable detour (Umweg): that it describes 'itself,' writes and unwrites". (P, p. 269) This entails that theses such as "death is the result or end of life" cannot be ascribed to Freud just like that. Neither can the Nietzschean tag of the "eternal recurrence of the same" apply to Freudian metapsychology.

Freud even turns into the devil, or at least he is the "devil's advocate", and this devilish turn explains his constant shifting between theory and autobiographical writing (P, p. 271).

A hundred pages later, Derrida seems to have forgotten the methodological prudence he had exhibited so far. Reaching the same passage at the end of his commentary, Derrida identifies Freud with Schopenhauer and with Heidegger, as if the equation between Freudian deconstruction (Abbau) and the death drive would be too much to bear. However, an equation between "deconstruction" and death would not bother Freud in the least, for at this point of his speculation, he sees that he has come too close to the dualistic theory of Schopenhauer who presented death as the purpose of human life. We have to read Freud carefully:

"We have unwittingly (unversehens, which means both "unexpectedly" and "without being fully aware"*)* steered our course into the harbour of Schopenhauer's philosophy" (BPP, p. 59, *Jenseits*, p. 259).

One should not remain in this safe but dead end:

"Let us make a bold attempt at another step forward" (BPP, p. 60).

This "bolder" step leads him to assume that libido or love can "rejuvenate" certain cells. Too much narcissism (seen here as the opposite of love for another being) generates death: hence, the uncontrolled reduplication of cells we find in cancer. Cancer displays the paradox of a disease brought about by a refusal to die, which thus destroys the organism. We now understand better why the ego drive can be equated with death, while the sexual drive can be equated with a life-giving force. This is clearer in the original: "Wir sind ja vielmehr von einer scharfen Scheidung zwischen Ichtrieben = Todestrieben und Sexualtrieben = Lebenstrieben ausgegangen". (*Jenseits*, p. 261).[6] Even if Freud is not explicitly saying that more love could cure cancer, he is not that far from such a thought. The couple of ego drive and death drive is opposed to the second couple of sexual drive and life. If we can identify here the seeds of the dualism of Eros versus Thanatos, we may note—a point that will not be lost on Lacan—that it is the Ego that is placed on the side of Thanatos.

The context of Freud's reversal when adding the last chapter to *Beyond the Pleasure Principle* was his decision to be treated for a Steinach operation that might "rejuvenate" him. In November 1924, Freud had a vasectomy performed in order to be "rejuvenated", as Steinach said when he promoted this operation as a male enhancement surgery. When the same vasoligation was performed on Yeats in 1936, Yeats reported an increased vitality in his sexual life and poetic creativity. Freud was less enthusiastic, although he believed that the operation had brought about a respite from his cancer.

A similar note was sounded by Derrida in what has been called his last interview given after he was told he had a few weeks to live due to his pancreatic cancer. As he quips, he would be happy with the idea of a simple resurrection, soul and body together, something he might look for. In another interview with Jean-Luc Nancy and Philippe Lacoue-Labarthe from the same period, Derrida returns to the moment when he made the terribly necessary decision to destroy his correspondence with Sylvie Agacinsky, as we now know while keeping absolutely everything, all the notes, post-its, and documents, which included all the letters he received in his life. The paradox is that in order to keep them safely, he had to deposit them in institutions like French archives, Irvine, or now Princeton, which prevented him from having easy access to them (including my letters to him, that people I know can read in Princeton or Irvine, to my dismay). We are back to the Freudian paradox of a "mystic writing pad" emblematizing memory and forgetting while pointing to the Unconscious as a hidden archive. This sends us back to Kafka's paradox as presented in one of his last stories, "The Burrow" (Der Bau). An animal looking like a mole or a badger rejoices over its hoarded goods in the numerous tunnels underground, but in the end, hearing unnamed enemies digging beneath his construction, feels safer by staying outside the burrow. The complexities of the animal's situation resemble those of deconstruction in its constant rephrasing of "construction" as "deconstruction" in order to avoid "destruction". As Stanley Corngold writes in a footnote explaining his translation

of a title given the text posthumously by Max Brod, a title that is itself impossible to render in English: "Kafka's title *Der Bau*, a word that recurs throughout his work, poses an insuperable difficulty for the translator: the German word means "building, construction", but since in the story we are obviously dealing with an underground construction produced by a badgerlike animal, the word of choice would seem to be "burrow". Once we have committed ourselves to this "burrow", however, we have sacrificed an element of tension that is a constitutive part of the story: the doubleness of a structure that is built horizontally underground but is represented in the mind of the badger-narrator in "higher" terms, terms more suitable to a structure built vertically underground" (Kafka 2007).

I do not need to pursue the analogy, it imposes itself: the legacy of Freud to deconstruction and the legacy of Derrida to psychoanalysis both function like this structure that is simultaneously underground and overground, rebuilt and unbuilt, abandoned as obsolete and dangerous only to be reinhabited later. It is not a coincidence that Kafka's story should remain unfinished, the manuscript ending with the deictic "the": "… but everything remained unchanged, the * * *" (Ibid., p. 189). The "burrow" is a "borrow" for such a debt will never be repaid.

**Funding:** This research receives no funding.

**Conflicts of Interest:** The author declares no conflict of interest.

## Notes

[1] Joseph Bernstein mentions the fact that "the Instagram account freud.intensifies has more than a million followers and posts memes like a portrait of Freud overlaid with the text "Every time you call your boyfriend 'Daddy', Sigmund Freud's ghost becomes a little stronger". See (Bernstein 2023).

[2] I want to pay homage to the memory of Philippe Sollers whose influence had been enormous in attempting a mediation during a crucial fight between Lacan and Derrida.

[3] Note 44 in (Derrida 1982) This note sketches the program Derrida followed in 1975 when debunking Lacan's "phonocentism" and Hegelian idealism in (Derrida 1987a). See the entire file of the numerous readings of (Muller and Richardson 1988) Here abbreviated as PP.

[4] (Poe 1983) Poe quotes Novalis's *Morale Ansichten*. The first sentence can be rendered as: "There are series of ideal occurrences that run parallel to Reality".

[5] "Robert (*impatiently*): No man ever lived on this earth who did not long to possess—I mean to possess in the flesh—the woman whom he loves. It is nature's law.//Richard (*contemptuously*): What is that to me? Did I vote it?" (Joyce 1992).

[6] "Our argument had as its point of departure a sharp distinction between ego-instincts, which we equated with death-instincts, and sexual instincts, which we equated with life-instincts" (BPP, p. 63).

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
