# Peer review of "Is a Purloined Letter Just Writing? Burrowing in the Lacan-Derrida Archive"

_humanities, doi:10.3390/h12060146_

Round 1
Reviewer 1 Report
Comments and Suggestions for Authors
Comments on “Is a Purloined Letter just Writing?”
General Comments:
· Generally, this is a strong essay summing up the Lacan/Derrida debate about letters and their destinations. The author clearly has interesting personal connections and recollections of the context in which this debate emerged (which scholars like myself, who are younger and from the US, would not have access to). The debate is generally explained clearly, with a good sense of the terms of the disagreement and why this conflict mattered.
· The thing that most feels like it’s missing is a clear, specific sense of what this essay adds to the debate. Particularly since this is such a long running debate and it has generated so many subsequent commentaries, analyses, and responses, it’s difficult for non-experts in this debate (i.e., like the readers of Humanities) to know clearly what is new in this essay versus what are existing/established stances in the debate. In short, I would like to see a more explicit statement about what this essay’s original intervention in the letter-debate is, and why it’s important for readers in 2023/2024 (whenever the essay is published). I get the sense that the author is so thoroughly versed in this debate that they are approaching the subject as though readers will be equally knowledgeable.
o On a related note, Earlie’s book seems important, but we never get a clear sense of what his argument is or how it changes the understanding of this debate. The essay might generally benefit from more direct, though still concise, summaries of key works discussed for the benefit of readers not super familiar with these philosophical/psychoanalytic positions.
· Abstract: What feels like it’s missing here is any sense of why this essay matters. This is connected to the problem I identified above. Show the stakes in the abstract so readers know why this essay would be worth reading.
· This is a minor thing and can easily be fixed in copy-editing, but the typeface and font size in the notes varies significantly, even within the same note sometimes.
Specific/Line Comments:
· Abstract Line 4: What is the book? I’d name it, with the author and title.
· Abstract Line 5: “Its” is a vague pronoun here. Does this refer to the controversy between Lacan and Derrida or to the recent book.
· Line 12: Consider giving a bit of an introduction to Earlie’s book. This seems important, but for at least the first four pages it goes undiscussed. At the least, maybe give us Earlie’s thesis/major intervention. Or, if you’re not going to do this, I would signpost that the book will be discussed in more detail after the background on the Lacan/Derrida clash.
o Then, since you do rely on Earlie’s book later in the text (e.g., the last paragraph on page 8, beginning at line 306), we do somewhere need a clear summary of the argument and its significance.
· Line 19: I’d put either a semi-colon or a full stop period between “prove one’s allegiance” and “but in fact.”
· Line 37: if the question mark after “violently” is meant to be there, I’d put it in parenthesis to suggest editorial commentary.
· Line 53: I think this sentence should read “then ultimately is a sign,” not “as a sign.”
· Line 147: This should read ‘an “Unconscious Grammatology”.’
· Line 189: Should this read “when Kafka asks his father to intervene”?
· Lines 193-203: If this is all Derrida’s text—which is what it seems like—this should be properly formatted as a block quote: the entire passage indented.
· Lines 211-215: The same as previous note, Kafka’s text should be set up as a block quote.
· Lines 225-231: Same note.
· Lines 236-248: Same note.
· Lines 284-296: Same note.
· Line 325: There’s an extra indent at the beginning of this paragraph.
· Lines 337-342: Block quote.
· Line 345: I think this should read “previous commentaries on Beyond the Pleasure Principle.”
· Line 439: Given that the word terrible is highlighted here and that the question mark immediately follows necessary, I think there are elements of editorial commentary here. The simplest way I would imagine this could be cleaned up is: “he made the terrible/necessary(?) decision.”
· Line 449: I don’t think the question mark appended to beneath is necessary as editorial commentary, so it may be a typo. If it is editorial commentary, I’m not sure what the comment signifies, what uncertainty it marks.
· Lines 454-461: This should be in a block quote because of the length of the passage. Because it should be block quoted, keeping the double quotation marks from the original is fine because the block quote itself would be without quotation marks.
Comments on the Quality of English LanguageThere are a couple of minor things, noted in the attached file.
Author Response
I have added a new introduction that tries to contextualize my readings in terms both of pedagogical interest and today's trends: the recent "return" of psychoanalysis under its Lacanian mode, and the comeback of deconstruction after the "woke" controversies have genrated a number of confusions. I attempt to untangle one crucial identification, Derrida's textuality and Lacan's analysis of the Letter in the Unconscious.

Reviewer 2 Report
Comments and Suggestions for Authors
This is a very well written discussion of the debate between Derrida and Lacan over Poe's story. I especially appreciate the author's addition of Derrida's discussion of Kafka's letter to his father to this debate. I appreciate as well the author's mature, level-headed writing style. Conceptually, the argument is sound and well-researched. The author is well versed in Lacanian theory and deconstruction; I am confident that others in this field will find their thoughts on how Derrida's analysis of Kafka's letter complicates his critique of Lacan interesting. The MS itself needs to be proofread, as it contains some typos, stylistic errors, and inconsistencies. Starting with page 1, line 12, the footnote number is italicized and there is a missing comma after the book title. The footnotes themselves need to be formatted properly. Page 2, line 53, as should be is. Line 65 contains an extra space. Page 3, line 82, the comma is misplaced. The parenthetical citations are formatted inconsistently (see the two atop page 9), as are block quotes (some are set off by a line break, others aren't, and none are indented). Page 6, line 264, he should be the. Page 7, lines 270-1, double dashes should be em dashes. Line 296 is missing a closed parenthesis. Line 303 has an extra space. I am sure another close-read would reveal more little errors. Once all those are cleaned up, this essay will be ready for publication.
Author Response
I am a terrible proof-reader and bad copy-editor but I have tried to revise the footnotes and the indentations in a more coherent manner.
